# Body Condition in the Tawny Owl *Strix aluco* near the Northern Limit of Its Range: Effects of Individual Characteristics and Environmental Conditions

**DOI:** 10.3390/ani12202843

**Published:** 2022-10-19

**Authors:** Tapio Solonen

**Affiliations:** Luontotutkimus Solonen Oy, Neitsytsaarentie 7b B 147, FI-00960 Helsinki, Finland; tapio.solonen@pp.inet.fi

**Keywords:** colour morph, frost seesaw, Julian date, snow cover, vole index, winter temperature

## Abstract

**Simple Summary:**

The present study examines how the variations in food supply and winter weather are reflected in the body condition of female and male Tawny Owls *Strix aluco* of different colour morphs in a population near the southern coast of Finland. Winter weather conditions before breeding seemed to have effects on the food availability of Tawny Owls: the depth of the snow cover showed a positive relationship, and the frequency of temperature fluctuations around the freezing point had a negative relationship. In females, intrinsic factors such as colour morph and age, as well as the body condition of the mate and the stage of the season, governed body condition. In males, only age and the stage of the season suggested associations with body condition. Probably due to the efficient use of alternative prey, the effects of fluctuations of vole populations on the body condition of Tawny Owls are only moderate.

**Abstract:**

The body condition of boreal species of vole-eaters seems to vary largely according to fluctuations in vole populations and weather conditions of the preceding winter. I studied females and males of the Tawny Owl *Strix aluco* of temperate origin near the northern limit of the species’ range in southern Finland to reveal if they show similar patterns to the boreal species. Winter weather conditions before breeding seemed to have pronounced effects on the food availability of Tawny Owls. In females, intrinsic factors such as colour morph and age, as well as the body condition of the mate and the stage of the season (Julian date), governed body condition. In males, only age and Julian date showed pronounced relationships with body condition. The results suggest that deep snow cover protects vole populations through winter until spring better than a minor amount of snow and that frequent temperature fluctuations around the freezing point in early spring make voles more available for owls that are preparing for breeding. This was also reflected positively in the body condition of female owls. Probably due to the efficient use of alternative prey, the effects of fluctuating vole populations on the body condition of Tawny Owls are, in general, only moderate.

## 1. Introduction

Good body condition, a favourable physiological state in terms of protein and fat [1,2], is needed for successful reproduction in birds [3]. In capital breeders [4,5], sufficient body condition for breeding is determined by the environmental conditions preceding the onset of egg-laying [6,7]. In birds of prey, large fat reserves are beneficial for females that defend the nest, while relatively small fat reserves should improve the agility of foraging males [8]. The initiation of egg-laying is determined by the foraging capacity of the male [9].

The Tawny Owl *Strix aluco* is a polymorphic, stationary, nocturnal, generalist predator, occupying a relatively continuous range across most of Europe from the Mediterranean to the Nordic countries [10,11]. Tawny Owl populations are thus confronted with widely varying spatial and temporal environmental conditions from the Mediterranean through temperate to harsh boreal ones [12,13,14,15,16]. Populations seem to have prominent differences in their post-glacial origin, as well as in their genetic structure [17,18]. This suggests that the reproduction of different populations is affected by different combinations of intrinsic and extrinsic factors. Reproduction near the northern limit of the species’ range is particularly affected by the highly fluctuating abundance of small voles, the staple prey of owls [19,20,21,22,23]. The plumage colour seems to signal fitness in survival and reproduction [12,24,25,26,27]. Grey Tawny Owls seemed to be more viable and productive than brown ones in cold environmental conditions [28,29].

The present study extends and summarises the results of some earlier studies [9,27,30]. It examines how the annual and seasonal variations in food supply and winter weather conditions are reflected in the body condition of breeding females and males of different colour morphs (grey and brown) in a population of Tawny Owls near the boreal southern coast of Finland. This population seems to live in somewhat different environmental conditions than populations studied elsewhere in Finland [27]. In this population, body condition seemed to be largely determined by the bird’s age, time of the season, and stage of the nestling period [30]. The female condition is positively affected by both male condition and female age, while the male condition is positively related to the age of males. These intrinsic factors seemed to outweigh the effects of the prominent extrinsic factors (general food supply and winter weather conditions). However, in an earlier study, the extrinsic factors seemed to be inadequately quantified when winter weather conditions were characterised only using midwinter temperature [9]. Therefore, besides including a diverse set of intrinsic factors in the analyses as explanatory variables, the present study aims to evaluate some weather variables that might characterise those environmental conditions where the studied population lives before the breeding season more accurately than evaluating the midwinter temperature alone.

Winter weather conditions may affect owls directly by increasing or decreasing body conditions according to fluctuating energy demands due to varying ambient temperatures (see [31]). In addition, they may have short-term or long-term effects on owls by changing the availability of food (see [32]). It is reasonable to believe that winter weather conditions do not necessarily have direct effects on the breeding season body condition and reproduction of owls. However, indirect effects are possible via the effects on food availability before the breeding season. This study focuses on the following predictions: (1) Winter weather conditions, in particular the amount of snow and the frequency of temperature fluctuations around the freezing point, affect the pre-breeding food supply of Tawny Owls (measured using the annual mean clutch size of the population [33]). (2) Weather conditions and weather-related effects on pre-breeding food supply may be reflected in the body condition of breeding owls. (3) Pre-breeding food supply modified by the stage of the season (Julian date) reflects in the female body condition via the body condition of the male.

## 2. Materials and Methods

### 2.1. Study Population and Field Work

I collected the present data on breeding Tawny Owls in an area of about 500 km^2^ on the southern coast of Finland (60° N, 25° E) (Appendix A) between 1986 and 2018 [27]. Up to about 300 nest boxes were available for owls relatively evenly throughout the study.

Around the middle of the nestling period, I captured female and male owls of breeding pairs at nests using a hoop net or trap for individual recognition, ringing, and examining their plumage and other body characteristics. Nests found with eggs were rechecked within the next three weeks to confirm the final clutch size. Brood size was the number of nestlings at the age of about 3–4 weeks. The total data consisted of 332 nestings of pairs that included 210 individual females and 205 individual males [27].

### 2.2. Characteristics of Breeding Birds

I sexed the owls based on their general size, wing length, body mass, and the incubation patch of females [9,34,35]. In classifying plumage colour, I divided the birds into two groups—grey and brown ones [27]. I determined the age of birds using the plumage patterns as 1 yr old, 2 yrs old, and 3 yrs old or older [36]. The maximum wing length [37] (indicating the size of the individual) and the body mass (measured using Pesola spring balance to the nearest gram) were used for estimating the indices of body condition [9] (Appendix A).

### 2.3. Extrinsic Factors

The essential requirement for successful reproduction, the availability of nesting sites, was fulfilled by providing a sufficient amount of nest boxes suitable for Tawny Owls. Relationships of various other extrinsic factors potentially affecting body condition and reproduction in Tawny Owls are visualised in Figure 1. I characterised general weather conditions and the severity of the preceding winter using local average winter temperatures, including January, February, and March, and the frequency of temperature fluctuations around the freezing point (“frost seesaw” [32]) in March, as well as the depth of the snow cover in the middle of March, measured at the Helsinki–Vantaa airport (60.33° N, 24.96° E) [38], roughly in the middle of the present study area. I monitored annual fluctuations in food supply indirectly using the annual average clutch size of the Tawny Owl population [34] (Appendix A). During breeding, the extrinsic explanatory factor which governed clutch size and body condition was the stage of the season (Julian date) (reflecting changes in food supply during the season).

### 2.4. Statistics

I examined relationships between the explanatory variables and dependent variables with generalised linear models (glm, family Gaussian) using the statistical package nlme in R version 4.1.2 (1 November 2021) [39,40]. I checked the normality of residuals using the Shapiro–Wilk test. In the analyses, the dependent variable was the food supply (vole index) or body condition of owls (condition index). Explanatory variables included intrinsic factors (colour morph, age, body condition) and extrinsic factors (mate body condition, Julian date, food supply, winter temperature, depth of the snow cover, and frequency of temperature fluctuations around the freezing point). Primary data are available in Appendix A.

## 3. Results

Winter weather conditions before breeding seemed to have effects on the food availability of Tawny Owls (Table 1). The effect of the general winter temperature was, however, minor. The depth of the snow cover showed a positive relationship, explaining 0.6%, while the frequency of the frost seesaw showed a negative relationship, explaining 9.2% of the variation in the vole index (food supply).

The body condition of Tawny Owls was better in females than in males (Table 2). Frost seesaw was the only weather variable that showed a discernible relationship with body condition: a minor positive relationship in females. In males, the respective relationship was even weaker. When intrinsic variables were also included, the direct effects of food and weather factors on body condition seemed minor (Table 3 and Table 4). In females, intrinsic factors such as colour morph and age, as well as mate condition and Julian date, governed body condition (Table 3). The body condition of brown females was better than that of grey females (0.195 vs. 0.177, *t* = 2.14, df = 330, *p* = 0.033). The mate body condition (positive relationship) explained 10.0% of the variation in the body condition of females (Figure 2). The Julian date (negative relationship) explained 17.9% of the variation in female body conditions (Figure 3). The contribution of the Julian date was 26.3% and 11.6% in grey and brown females, respectively. In males, only age and Julian date showed clear associations with body conditions (Table 4). The Julian date (negative relationship) explained 4.2% of the variation in the male body condition (Figure 4). The contribution of Julian date was 3.7% and 5.2% in grey and brown males, respectively.

## 4. Discussion

The present results suggest that winter weather conditions may considerably affect the food availability of Tawny Owls before breeding. These relationships between the winter weather variables and the vole abundance index suggest that a deep snow cover protects vole populations through winter until spring better than a minor amount of snow (see [41]). On the other hand, frequent temperature fluctuations around the freezing point (frost seesaw) in early spring make voles more available for owls that are preparing for breeding. This was also reflected positively in the body condition of female owls. During earlier periods of winter, however, the consequences of frequent frost seesaws were detrimental for both voles and owls [32,42]. It can be expected that the frequency of such circumstances will increase due to ongoing climate warming.

The general vole index [33] used here, in fact, describes the average general food availability of the owl population at the beginning of breeding. The index value mainly characterises fluctuations in the availability of the staple food of owls, i.e., small voles. However, it also includes the alternative prey that dampens fluctuations in the total food supply, as well as modifies the effects of environmental factors (such as weather conditions) on the general availability of food. Pre-breeding food supply, modified later by the circumstances of the stage of the season (indicated by the Julian date), is reflected in the female body condition via the male condition. Weak associations between the vole index and the body condition indices stem from the time difference between the indices: the vole index characterises the situation in the pre-breeding period, while the condition index considers the situation about six weeks later during the second half of the nestling period.

The body condition of brown females was better than that of grey ones. This suggests that the environmental circumstances favoured the brown colour morph (cf. [12,29]). Old birds bred early in the season and were in better condition than young ones that bred later, suggesting the importance of experience.

As all capital breeders, Tawny Owls should exhibit good body condition before breeding. In my study area, the most important prey species responsible for this at the beginning of the breeding season is the Field Vole *Microtus agrestis* [20]. Following the melting of the snow cover, the improved availability of European Water Voles *Arvicola amphibius* is of decisive importance [20,30].

From the individual point of view, the seasonal decline in the food supply means that the body condition of birds decreases, and it is increasingly challenging for the male to fatten the female to the breeding condition. From the population point of view, this means that breeding shifts later, and the later clutches are smaller than the earlier ones [43]. The apparently minor effect of various extrinsic factors on reproduction probably results from the fact that during unfavourable environmental conditions, owls breed only in the best territories and only the individuals in the best body condition are able to breed [20,22,44].

## 5. Conclusions

Winter weather conditions before breeding affect the food availability of Tawny Owls in various ways. For instance, a deep snow cover protects voles, their staple prey, through winter until spring better than a minor amount of snow, while frequent temperature fluctuations around the freezing point in early spring make voles readily available for owls. Local environmental conditions, particularly local prey supply, and the individual characteristics of males are strongly involved in determining the body condition of Tawny Owls. However, the body condition is relatively resistant to variable foraging conditions due to alternative prey, as could be expected for a generalist predator of southern origin. Thus, due to the efficient use of alternative prey, the effects of fluctuating vole populations on the body condition of the Tawny Owl are, in general, moderate.

## Figures and Tables

**Figure 1 animals-12-02843-f001:**
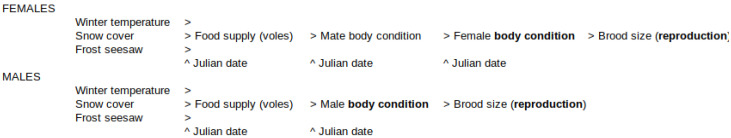
Path diagrams of the relationships of some extrinsic factors affecting body condition and reproduction in female and male Tawny Owls *Strix aluco* in southern Finland (see Section 2.3). The symbols > and ^ indicate the direction of the effect.

**Figure 2 animals-12-02843-f002:**
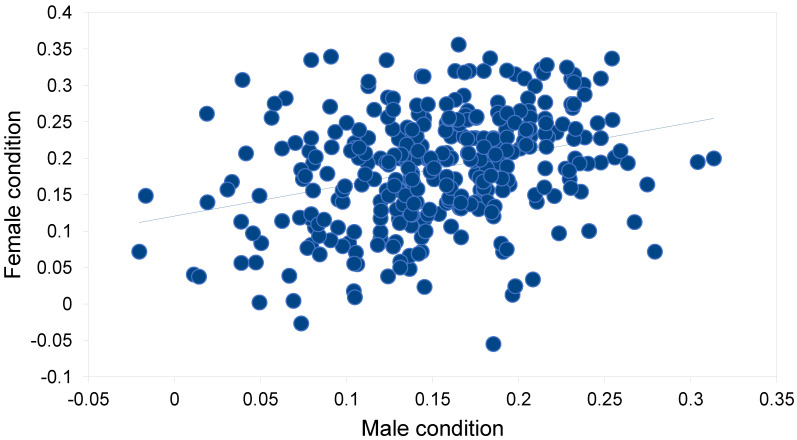
Relationship between the male body condition and female body condition of breeding pairs of Tawny Owls *Strix aluco* in the studied population in southern Finland (f(x) = 0.428x + 0.121, *r* = 0.316, df = 330, *p* < 0.001).

**Figure 3 animals-12-02843-f003:**
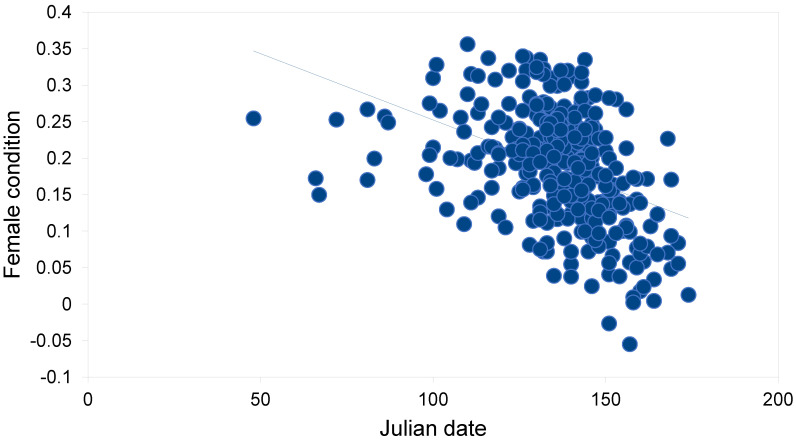
Relationship between Julian date and female body condition of breeding Tawny Owls *Strix aluco* in the studied population in southern Finland (f(x) = −0.002x + 0.434, *r* = −0.424, df = 330, *p* < 0.001).

**Figure 4 animals-12-02843-f004:**
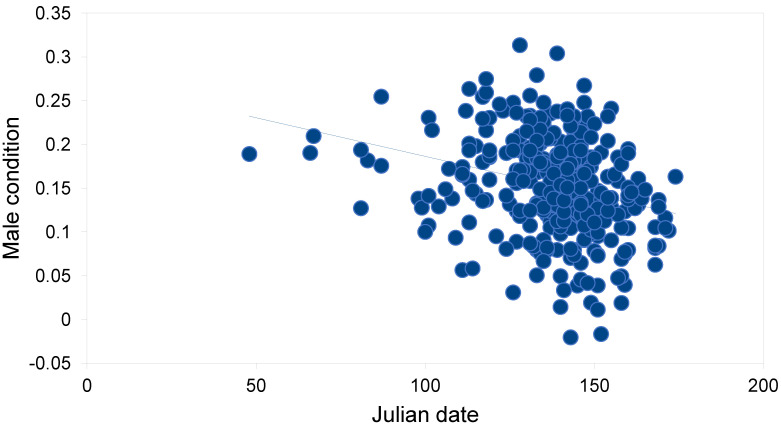
Relationship between Julian date and male body condition of breeding Tawny Owls *Strix aluco* in the studied population in southern Finland (f(x) = −0.001x + 0.275, *r* = −0.275, df = 330, *p* < 0.001).

**Table 1 animals-12-02843-t001:** Relationships between the winter weather variables examined and the general vole abundance index (annual mean clutch size of the Tawny Owl *Strix aluco* population studied). Generalised linear model (glm): Voleindex.model <- glm (vole index ~ winter temperature + snow cover + frost seesaw). The parameter estimates of data (mean and 95% confidence interval) are given.

Variable	Value	SE	df	*t*	*p*	Mean (95% CI)	AIC	Adj R^2^
(Intercept)	4.7170	0.1535	300	30.7	<0.001	3.81 (3.75–3.86)		
Winter temp.	0.0555	0.0193	28	2.88	0.004	−3.70 (−3.90–−3.49)	500.6	−0.002
Snow cover	0.0073	0.0022	28	3.30	0.001	18.40 (16.67–20.12)	498.1	0.006
Frost seesaw	−0.0516	0.0078	28	−6.61	<0.001	16.29 (15.90–16.68)	467.9	0.092

**Table 2 animals-12-02843-t002:** Relationships between the winter weather variables examined and the body condition of female and male Tawny Owls *Strix aluco* near the southern coast of Finland: generalised linear models (glm). The parameter estimates of data (mean and 95% confidence interval) are given.

Variable	Value	SE	df	*t*	*p*	Mean (95% CI)	AIC	Adj R^2^
*Female condition*								
(Intercept)	0.1274	0.0244	300	5.21	<0.001	0.186 (0.178–0.194)		
Winter temp.	−0.0044	0.0031	28	−1.45	0.148	−3.70 (−3.90–−3.49)	−756.7	−0.003
Snow cover	−0.0005	0.0004	28	−1.42	0.155	18.40 (16.67–20.12)	−757.0	−0.002
Frost seesaw	0.0032	0.0012	28	2.54	0.012	16.29 (15.90–16.68)	−761.4	0.011
*Male condition*								
(Intercept)	0.1246	0.0181	300	6.87	<0.001	0.153 (0.147–0.159)		
Winter temp.	−0.0007	0.0023	28	−0.30	0.768	−3.70 (−3.90–−3.49)	−958.6	−0.003
Snow cover	0.0000	0.0003	28	0.033	0.974	18.40 (16.67–20.12)	−958.7	−0.003
Frost seesaw	0.0016	0.0009	28	1.706	0.089	16.29 (15.90–16.68)	−961.5	0.006

**Table 3 animals-12-02843-t003:** Relationships between some intrinsic and extrinsic factors and the body condition of female Tawny Owls *Strix aluco* near the southern coast of Finland: generalised linear model (glm). The parameter estimates of data (mean and 95% confidence interval) are given.

Variable	Value	SE	df	*t*	*p*	Mean (95% CI)	AIC	Adj R^2^
(Intercept)	0.2975	0.0677	294	4.39	<0.001	0.186 (0.178–0.194)		
Female colour	0.0193	0.0073	294	2.63	0.009	1.527 (1.473–1.581)	−761.3	0.011
Female age	0.0036	0.0011	294	3.11	0.002	4.136 (3.780–4.491)	−768.4	0.032
Mate colour	0.0200	0.0073	294	2.72	0.007	1.500 (1.446–1.554)	−761.0	0.010
Mate age	−0.0003	0.0015	294	−0.20	0.842	3.831 (3.556–4.106)	−757.3	−0.001
Mate condition	0.2507	0.0669	294	3.75	<0.001	0.153 (0.147–0.159)	−791.5	0.097
Julian date	−0.0018	0.0002	294	−7.63	<0.001	136.4 (134.5–138.4)	−822.3	0.177
Vole index	−0.0063	0.0083	27	−0.76	0.450	3.806 (3.751–3.861)	−757.3	−0.001
Winter temp.	−0.0052	0.0027	27	−1.96	0.051	−3.70 (−3.90–−3.49)	−756.7	−0.003
Snow cover	−0.0001	0.0003	27	−0.32	0.746	18.40 (16.67–20.12)	−757.0	−0.002
Frost seesaw	0.0015	0.0012	27	1.28	0.202	16.28 (15.90–16.68)	−761.4	0.011

**Table 4 animals-12-02843-t004:** Relationships between some intrinsic and extrinsic factors and the body condition of male Tawny Owls *Strix aluco* near the southern coast of Finland: generalised linear model (glm). The parameter estimates of data (mean and 95% confidence interval) are given.

Variable	Value	SE	df	*t*	*p*	Mean (95% CI)	AIC	Adj R^2^
(Intercept)	0.2382	0.0546	295	4.36	<0.001	0.153 (0.147–0.159)		
Male colour	0.0087	0.0060	295	1.45	0.149	1.500 (1.446–1.554)	−960.1	0.001
Male age	0.0026	0.0012	295	2.17	0.031	3.831 (3.556–4.106)	−965.6	0.018
Mate colour	0.0107	0.0060	295	1.78	0.076	1.527 (1.473–1.581)	−960.7	0.003
Mate age	0.0017	0.0009	295	1.86	0.063	4.136 (3.780–4.491)	−965.3	0.017
Julian date	−0.0010	0.0002	295	−5.23	<0.001	137.9 (136.0–139.8)	−984.8	0.073
Vole index	−0.0054	0.0068	27	−0.80	0.426	3.806 (3.751–3.861)	−958.8	−0.002
Winter temp.	−0.0016	0.0022	27	−0.74	0.463	−3.70 (−3.90–−3.49)	−958.6	−0.003
Snow cover	0.0002	0.0003	27	0.70	0.487	18.40 (16.67–20.12)	−958.7	−0.003
Frost seesaw	0.0009	0.0010	27	0.91	0.366	16.28 (15.90–16.68)	−961.5	0.006

## Data Availability

Data are available in Appendix A.

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
