# Peer review of "Body Condition in the Tawny Owl Strix aluco near the Northern Limit of Its Range: Effects of Individual Characteristics and Environmental Conditions"

_animals, 2022, doi:10.3390/ani12202843_

Round 1
Reviewer 1 Report (Previous Reviewer 2)
The author added in the revision confidence intervals for parameters used in the GLM analyses. However, this info is not useful as it does not provide direct information. What would improve the quality of the paper is adding parameter estimates of data, e.g. averages with confidence intervals of parameters such as clutch size, brood size, body weight, etc. This is currently lacking (at least I did not find any data except the raw data in the supplemental table). I leave it to the author to add it or not.
Author Response
Please see the attachment.

Reviewer 2 Report (Previous Reviewer 3)
Dear Author, I am completely satisfied with the review done.
Author Response
Please see the attachment.

This manuscript is a resubmission of an earlier submission. The following is a list of the peer review reports and author responses from that submission.
Round 1
Reviewer 1 Report
Dear Author,
I have only comments to statistical analysis section. In my opinion data analysis must by improved. I put my comments directy to MS.
Best wishes

Reviewer 2 Report
This study aims at assessing the influence of environmental conditions on body condition and reproduction of tawny owls. The manuscript is generally well written (except the use of passive voice). I however have two major concerns. First, the methods are not sufficiently clearly described, I do not really know how you assessed body condition and I completely missed the description of assessing vole abundance. Second, you present most of the data only indirectly by an excessive amount of significance testing. What I miss are the actual data, for example parameter estimates of reproduction and even body condition. I propose to reduce the many tables with test results to one or two and instead present tables with averages and confidence intervals of various parameters for condition and reproduction.
Minor comments:
Line 43. This publication should be of interest for you: Barnett, C. A., Suzuki, T. N., Sakaluk, S. K. and Thompson, C. F. (2015) Mass-based condition measures and their relationship with fitness: in what condition is condition? Journal of Zoology 296: 1–5
Lines 101-110. Study area description is usually in present tense as the conditions are probably still valid.
Lines 114-115. Unclear at which moment you recorded clutch size and number of fledglings, did you both at a time for which you are certain you counted all?
Lines 119-124. The test results are here of little value, better provide parameter estimates or regression equations.
Lines 127-129. Use the first person here and throughout the manuscript when you refer to yourself instead of the passive voice.
Line 136-146. Not clear how you calculated body condition, the citation is not sufficient. Body mass to wing length? The description in this paragraph also needs clarifying, I could not follow the logic behind these plots. If body mass and wing length are correlated, then the quotient of them is not a good index for condition.
Fig. 1 . I do not see the value in this figure.
Line 162. If you use clutch size as proxy for food supply, better name it clutch size and not food supply. Otherwise you should measure actual food supply (e.g. vole abundance).
Line 177. Supplement. I miss the information on body mass and wing length in the excel file, it is more useful to provide raw data here, so that the reader gets more information.
Table 1. I could not find any descriptions of the methods to estimate vole abundance. This is necessary if you want to present results here. More useful than statistics would be presenting parameter estimates.
Line 187. Body condition is “better” rather than “higher”.
Fig. 5. Not clear here if brood size decreases with time (by mortality), which is rather trivial, or if brood sizes of clutches that were laid at different dates decreased. This needs clarification. If it is the first case, then the figure is not necessary as of course the brood size decreases in time by mortality.
Reviewer 3 Report
Manuscript ID: animals-1694833
Body Condition and Reproduction in the Tawny Owl Strix aluco near the Northern Limit of Its Range: Effects of Individual Characteristics and Environmental Conditions
By Tapio Solonen
Review
This is nicely written paper, let maybe Results section is too condensed. I read it with great interest, and have no criticism as for the data and analysis. However, there are few comments on formatting and presentation issues.
Simple summary
Italics not needed for this text.
Line 16: “point of the season” will be not understandable to readers
Line 17: food supply, expressed as vole index.
Abstract
Italics not needed for this text, except of species name
Line 30: remove “themselves”
Introduction
Line 54: please cite as [1,10–12]
Material and Methods
Map of the investigated area with locations of owl pairs would be nice as Figure 1, not all reads will refer to the previous papers
Line 130–131: could this be rewritten in more readable way?
Line 162: “food supply” needs an explanation; vole index should be explained in some details, not only referred. Explanation should be added also to Supplement 1.
Lines 176–177: could this be cited as Table S1? Supplementary materials are perfect, but these excel Table should be formatted as tables, giving them caption and explanation about the variables presented. Presented material is of great value, it just needs some additional text.
Results
It is very concise presentation in the form of text. I personally would like to see at least some more sentences, explaining results – for each table and figure presented.
Table captions: they did not follow Template
Table body – italics not expected for all Table body text
Table 3,4,5,6 – it was said, that significant variables are in bold text. But they are not, at least not all.
Lines 211, 214: mistypes
Discussion
Lines 271, 275: italics for species name
Line 311: Myodes or Clethrionomys?
Back matter
Check Template, Back matter is presented not as required
Line 318: Table S1? See Template
References
DOI is not presented for some references
Volume should be in italics
In [33] doi is enough
Lines 400, 404, 414, 414 – additional spaces (or other symbols) are inserted?
Round 2
Reviewer 2 Report
The revision has improved the clarity of the methods. However, the reader now needs to read the supplements to know which methods you used. I would still rate the manuscript as having good data, whereas the presentation is much weaker. Significance testing is at best a small part of data analysis (I would always give priority to probability mathematics instead of significance testing, which rather is an artefact of the 20th century science, see e.g. Johnson, D.H. 1999. The insignificance of statistical significance testing. Journal of Wildlife Management 63: 763-772), to a good analysis parameter estimates are essential. However, data presentation is the decision and responsibility of the author, so I will not comment on it further, it is the decision of the editor if this meets the journal standards. I still encourage you to more directly present your data and have another thorough proof read of the manuscript.